# MicroRNAs as Predictors of Future Uterine Malignancy in Endometrial Hyperplasia without Atypia

**DOI:** 10.3390/jpm12020311

**Published:** 2022-02-18

**Authors:** Chiao-Yun Lin, Ren-Chin Wu, Lan-Yan Yang, Shih-Ming Jung, Shir-Hwa Ueng, Yun-Hsin Tang, Huei-Jean Huang, Hsiu-Jung Tung, Cheng-Tao Lin, Hsuan-Yu Chen, Angel Chao, Chyong-Huey Lai

**Affiliations:** 1Department of Obstetrics and Gynecology, College of Medicine, Chang Gung Memorial Hospital and Chang Gung University, Taoyuan 333, Taiwan; yunhsin.tang@gmail.com (Y.-H.T.); hueijean.huang@gmail.com (H.-J.H.); thjami@cgmh.org.tw (H.-J.T.); 51424@cgmh.org.tw (C.-T.L.); r91641001@gmail.com (H.-Y.C.); drangiechao@gmail.com (A.C.); 2Gynecologic Cancer Research Center, College of Medicine, Chang Gung Memorial Hospital and Chang Gung University, Taoyuan 333, Taiwan; renchin.wu@gmail.com (R.-C.W.); ming22@cgmh.org.tw (S.-M.J.); susie.ueng@gmail.com (S.-H.U.); 3Department of Pathology, College of Medicine, Chang Gung Memorial Hospital and Chang Gung University, Taoyuan 333, Taiwan; 4Biostatics Unit, Clinical Trial Center, College of Medicine, Chang Gung Memorial Hospital and Chang Gung University, Taoyuan 333, Taiwan; lyyang0111@gmail.com; 5Clinical Informatics and Medical Statistics Research Center, College of Medicine, Chang Gung University, Taoyuan 333, Taiwan

**Keywords:** endometrial hyperplasia, endometrial cancer, phosphatase and tensin homolog (PTEN), microRNA

## Abstract

The histological criteria for classifying endometrial hyperplasia (EH) are based on architectural crowding and nuclear atypia; however, diagnostic agreement among pathologists is poor. We investigated molecular biomarkers of endometrial cancer (EC) risk in women with simple hyperplasia or complex hyperplasia without atypia (SH/CH-nonA). Forty-nine patients with EC preceded by SH/CH-nonA were identified, of which 23 were excluded (15 with complex atypical hyperplasia (CAH), six not consenting, one with a diagnosis < 6 months prior, and one lost to follow-up). The EH tissues of these patients were compared with those of patients with SH/CH-nonA that did not progress to EC (control) through microRNA (miRNA) array analysis, and the results were verified in an expanded cohort through reverse transcription-quantitative polymerase chain reaction (RT-qPCR). MiRNA arrays analyses revealed 20 miRNAs that differed significantly (*p* < 0.05, fold change > 4) between the control (*n* = 12) and case (*n* = 6) patients. Multiplex RT-qPCR for the 20 miRNAs in the expanded cohort (94 control and 25 case patients) led to the validation of miR-30a-3p (*p* = 0.0009), miR-141 (*p* < 0.0001), miR-200a (*p* < 0.0001), and miR-200b (*p* < 0.0001) as relevant biomarkers, among which miR-141, miR-200a, and miR-200b regulate the expression of phosphatase and tensin homolog (PTEN). For the prediction of EC, the area under the curve for miR-30a-3p, miR-141, miR-200a, and miR-200b was 0.623, 0.754, 0.783, and 0.704, respectively. The percentage of complete PTEN loss was significantly higher in the case group than in the control group (24% vs. 0%, *p* < 0.001, Fisher’s exact test). A combination of complete PTEN loss and miR-200a provided optimal prediction performance (sensitivity = 0.760; specificity = 1.000; positive predictive value = 1.000; negative predictive value = 0.937; accuracy = 0.947). MiR-30a-3p, miR-141, miR-200a, miR-200b, and complete PTEN loss may be useful tissue biomarkers for predicting EC risk among patients with SH/CH-nonA.

## 1. Introduction 

Endometrial cancer (EC) is the leading gynecologic malignancy in industrialized countries, and the incidence rate of EC has increased over successive generations in several countries, many of which are undergoing rapid socioeconomic transitions [1]. In Taiwan, the incidence of EC has increased considerably in the past 30 years from less than 100 to more than 1000 cases per year, and it has been the leading gynecologic cancer in Taiwan since 2012 [2].

The World Health Organization categorizes endometrial hyperplasia (EH) as simple hyperplasia (SH), complex hyperplasia without atypia (CH-nonA), simple atypical hyperplasia, or complex atypical hyperplasia (CAH) on the basis of architectural crowding and nuclear atypia [3]. The severity of EH reflects the risk of EC. Kurman et al. reported that the risk of progression to EC for patients with SH, CH-nonA, and CAH is <1%, 3%, and 29%, respectively [4]. However, several clinical cases of patients with SH/CH-nonA progressing to EC within a short period have been reported. A major problem is that the diagnostic agreement pertaining to the subjective appraisal of hematoxylin and eosin staining for endometrial neoplasms is poor. The Gynecologic Oncology Group conducted a reproducibility study of 306 women with referring diagnoses of CAH. The overall kappa value for the independent pathologists panel diagnosis of CAH was 0.28. Specifically, 29% of conditions were diagnosed adenocarcinoma, 18% were diagnosed as SH/CH-nonA, and 7% were diagnosed as cycling endometrium [5].

Research on biomarkers for early cancer diagnosis is growing [6]. Because microRNAs (miRNAs) are small and well preserved in formalin, researchers are searching for miRNA biomarkers and the corresponding target genes indicative of transitions from SH/CH-nonA to EC over a long-term follow-up, miRNAs are evolutionarily conserved, non-coding RNAs that are usually between 21 and 25 nucleotides in length; they function by binding to the 3′untranslated regions of mRNAs, where they repress protein translation or promote mRNA degradation [7]. Evidence suggests that miRNAs have oncogenic potential, and aberrant miRNA expression has been detected in numerous types of human cancers [8,9,10]. For example, the miR-200 family (including miR-200a/b/c, miR-141, and miR-429) was reported to be upregulated in EC tissue as compared with CAH and normal tissue [11]. Two miRNA signatures can be used to classify EC significantly more effectively than a single signature can [12]. Donkers et al. listed miR-205, the miR-200 family, miR-135b, miR-182, miR-183, and miR-223 as potential biomarkers for EC diagnosis, and they highlighted cases of CAH that are prone to rapid progression to EC [13].

The profiles of biomarkers may be used to predict the potential for progression from EH to EC. Lee et al. [14] reported significantly higher expression of five miRNAs (i.e., miR-182, miR-183, miR-200a, miR-200c, and miR-205) in EC tissue than in CAH, SH, and normal endometrial tissue. Tantbirojn et al. [15] reported the absence of the phosphatase and tensin homolog (PTEN) protein in 60% of EC tissue samples and 24% of typical EH (SH/CH-nonA) tissue samples. Sangwan et al. [16] reported that the expression of cyclin D1 is significantly higher in EC than in SH. Russo et al. compared progressing EH (*n* = 15) with resolving EH (*n* = 17) tissue through next-generation sequencing and discovered that *PTEN*, *PIK3CA*, and *FGFR* variants were more frequent in progressing EH (60%, 60%, and 40% for *PTEN*, *PIK3CA*, and *FGFR*, respectively) than in resolving EH (35%, 12%, and 24% for *PTEN*, *PIK3CA*, and *FGFR*, respectively); they noted that no single gene could be used predict progression more accurately than pathological classification [17]. However, miRNA profiling can reveal early signs of future EC for patients with SH/CH-nonA, and few studies have thoroughly investigated this topic.

We used miRNA array analyses to compare the tissue levels of molecular biomarkers in women with an initial diagnosis of SH/CH-nonA between those who developed subsequent EC (case group) and those who did not (control group). We verified our results by using an expanded cohort and performing reverse-transcription-quantitative polymerase chain reaction (RT-qPCR). 

## 2. Materials and Methods

### 2.1. Tissue Samples

This study was approved by the Institutional Review Board of Chang Gung Memorial Hospital in Taiwan (IRB approval number: 100-4355A3 and 201700921B0). For the recruitment of case group participants, we identified patients with a history of endometrial SH/CH-nonA that subsequently developed into EC (>6 months from EH to EC) were identified by using our cancer database and invited them to participate; they were only enrolled after providing written informed consent. Forty-nine patients with EC preceded by SH/CH-nonA (case) were identified; among them, 23 were excluded: 15 had CAH, 6 declined to provide informed consent, 1 was diagnosed within 6 months, and 1 was lost to follow-up. Patients were prospectively enrolled in the control group if they had undergone hysteroscopic resection of endometrial lesions proven through subsequent pathology to be SH/CH-nonA and if progression to EC did not occur at least 6 years. The median time from EH to EC was 8.71 (range, 0.55–19.84) years (Appendix A). Two pathologists (RCW and SMJ) independently reviewed all histological slides, and any discrepancies resolved through consensus with a third pathologist (SHU). The patients’ clinical information was retrieved from the electronic medical records system of Chang Gung Memorial Hospital.

### 2.2. RNA Extraction from Formalin-Fixed Paraffin-Embedded Tissues

Five formalin-fixed paraffin-embedded (FFPE) blocks constituting 10-μm thick slices were selected and the paraffin was removed in deparaffinization solution (Qiagen, Hilden, Germany). RNA was extracted from the FFPE samples with the miRNeasy FFPE Kit (Qiagen) in accordance with the manufacturer’s instructions [18]. In brief, the samples were lysed through digestion with proteinase K at 56 °C for 15 min and treated with DNase at room temperature for 15 min. The RNA was purified through filter cartridges and quantified on a bioanalyzer (Agilent Technologies, Waldbronn, Germany).

### 2.3. MiRNA 3.0 Array Analyses and Target Prediction of miRNA

To investigate the differential expression of miRNAs between groups, Affymetrix miRNA 3.0 array (containing 2578 human mature miRNA probe sets; Affymetrix, Santa Clara, CA, USA) were used in accordance with the manufacturer’s instructions [19]. In brief, 1 μg of total RNA from each sample was subjected to a tailing reaction and labeled with the FlashTag RNA Labeling Kit (Genisphere, Hatfield, PA, USA); subsequently, a biotinylated signal molecule was ligated to the RNA sample in accordance with manufacturer’s instructions [20,21]. Each sample was then hybridized to the miRNA 3.0 Array at 48 °C for 16 h and then washed and stained on a Fluidics Station 450 (Affymetrix). After staining had been completed, each chip was scanned on a GeneChip Scanner 3000 7G (Affymetrix). The expression levels of the miRNA transcripts were determined with the probe set and Command Console 3.2 (Affymetrix). The freely downloadable Affymetrix Transcriptome Analysis Console Software was used to analyze the miRNA 3.0 Array. We used the miRBase database (http://www.mirbase.org, accessed on 5 April 2021) to identify potential target sites of miRNA.

### 2.4. Reverse Transcription

FFPE tissue RNA (350 ng) was reverse transcribed using a TaqMan MicroRNA Reverse Transcription Kit and TaqMan MicroRNA Assays (Applied Biosystems, Foster City, CA, USA) in accordance with the relevant version of the manufacturer’s protocol [10]. In brief, RT primers (the portion bound to the 3′ end of mature miRNAs) corresponding to the 21 miRNAs from the TaqMan MicroRNA Assays were mixed to convert the miRNAs into their corresponding cDNAs in a single reaction, and individual PCR was performed under the following cycling conditions: 16 °C for 30 min followed by 42 °C for 30 min and 85 °C for 5 min. The products were diluted with 0.1 × TE buffer before quantitative PCR.

### 2.5. Quantitative PCR

The miRNA levels in the FFPE tissues were analyzed with the TaqMan MicroRNA Assay, and normalized to endogenous control miR-16. The amplification conditions were as follows: initial denaturation for 10 min at 95 °C, followed by 45 cycles of 95 °C for 15 s and 60 °C for 1 min. The reactions were performed using an ABI PRISM 7900 HT (Applied Biosystems). The mean cycle threshold (Ct) value for each measurement was calculated. The analytical detection limit was defined as a Ct value of 40; thus, miRNAs with a Ct of >40 were regarded as undetectable.

### 2.6. Immunohistochemistry

FFPE tissue slices (4 μm thick) were deparaffinized in xylene and rehydrated in a series of graded ethanol baths. Immunohistochemistry (IHC) was performed as previously described [22]. Sections were then stained with anti-rabbit PTEN monoclonal antibody (Cell Signaling Technology, Danvers, MA, USA) or anti-mouse BECN1 antibody (Santa Cruz Biotechnology, CA, USA) by using an automated IHC stainer and Ventana Basic DAB Detection Kit (Tucson, AZ, USA). Counterstaining was performed with hematoxylin. For PTEN staining, individual glands were interpreted as PTEN-null if the epithelial cells were negative for both cytoplasmic and nuclear staining and the mesenchymal cells (used as an internal control) were positive [14]. Lesions were scored as PTEN-negative (complete PTEN loss) if >95% of glands were PTEN-null, as PTEN-positive if PTEN-null glands were absent, and as heterogeneous if PTEN-null glands accounted for <95% of the epithelium.

### 2.7. Statistical Analysis

A nonparametric Mann–Whitney U test was performed to compare continuous variables, and Fisher’s exact test was performed to assess the levels of PTEN and BECN1 in the hyperplastic tissue samples. The optimal cutoff values for the potential biomarkers were determined through analysis of the area under the reiver operating curve (AUC). The AUC and 95% confidence intervals (CIs) were computed to assess the discriminative value of miR-30a-3p, miR-141, miR-200a, and miR-200b as biomarkers. The sensitivity, specificity, positive predictive value (PPV), negative predictive value (NPV), and accuracy of the predicted probability values for the miRNAs and combined (miRNA and PTEN loss) models were calculated with the integrated “perfcurve” function of MATLAB (2015a). Statistical analyses were performed in SPSS (version 22, Chicago, IL, USA). All tests were two-sided, and *p* values of <0.05 were regarded as statistically significant.

## 3. Results

### 3.1. Use of miRNA Arrays to Measure miRNA Levels in Clinical Samples

We used miRNA microarrays to analyze the miRNA profiles of FFPE hyperplastic tissues. The case group (*n* = 6) comprised patients with a history of SH/CH-nonA that developed into EC after more than 6 months (from our cancer database). The control group (*n* = 12) comprised patients with SH/CH-nonA that did not progress to EC at least 6 years. The microarray results revealed 20 miRNAs that differed significantly between the control and case groups (*p <* 0.05, fold change > 4; Figure 1).

### 3.2. TaqMan RT-qPCR to Measure miRNA Expression in Expanded Cohort

We further analyzed the expression of the 20 miRNAs (miR-10a, miR-15a, miR-21, miR-25, miR-28-3p, miR-29a, miR-30a-3p (miR-30a*), miR-30b, miR-30e, miR-141, miR-146a, miR-146b-5p, miR-181d, miR-194, miR-200a, miR-200b, miR-451, miR-660, miR-3613-5p, and miR-4487) in an expanded cohort, using FFPE hyperplastic tissue samples and performing multiplex RT-qPCR analysis. The case group (*n* = 25) comprised patients identified from our cancer database who had a history of endometrial SH/CH-nonA that subsequently developed into EC within 6 months. The control group (*n* = 94) comprised patients with endometrial SH that did not progress to EC at least 6 years. Among the 20 miRNAs, miR-30a-3p (*p* = 0.0009), miR-141 (*p* < 0.0001), miR-200a (*p* < 0.0001), and miR-200b (*p* < 0.0001) exhibited significantly higher levels in the case group than in the control group (Figure 2).

Next, we examined the AUC, sensitivity, and specificity to evaluate the performance of the four miRNAs as biomarkers of EC risk among patients with SH/CH-nonA. For the discrimination, the AUC values for miR-30a-3p, miR-141, miR-200a, and miR-200b were 0.623 (95% CI, 0.4751–0.7708), 0.754 (95% CI, 0.6336–0.8738), 0.783 (95% CI, 0.655–0.91), and 0.704 (95% CI, 0.5575–0.851), respectively. The sensitivity, specificity, and accuracy of miR-30a-3p, miR-141, miR-200a, and miR-200b were 0.600, 0.674, and 0.658; 0.480, 0.966, and 0.860; 0.600, 1.000, and 0.912; and 0.560, 0.899, and 0.825, respectively (Figure 3 and Table 1).

### 3.3. BECN1 and PTEN in EH Tissue Samples and Their Association with miRNA Expression

Among our four potential biomarkers, miR-30a-3p was reported to modulate autophagy by targeting BECN1 [23], and the other three miRNAs (i.e., miR-141, miR-200a, and miR-200b) were reported to be correlated with pathological findings and PTEN expression in EC [24]. Therefore, we investigated the association between the aforementioned miRNAs and the status of target genes. Figure 4 shows the PTEN proteins that were detected through IHC in the tissue samples from the control and case groups. The percentage of complete PTEN loss was significantly higher in the case group than in the control group (24.0% vs. 0%, *p <* 0.001, Fisher’s exact test; Table 2). Furthermore, in the case group, the EC tissue exhibited significantly more complete PTEN loss than the EH tissue did (56.0% vs. 24.0%, *p* = 0.001), and the percentage of PTEN-positive EH tissue in the case group was significantly higher than that in the subsequent EC tissue (0% vs. 36%, *p* = 0.002; Fisher’s exact test; Table 2). However, when the histological scores of BECN1 in the FFPE tissues from the control and case patients were analyzed, and no differences were detected (Appendix A). 

### 3.4. The miRNA Expression in EC and EH Tissue of Case Group

We also compared the miRNA expression of the EC and EH tissues of the case group. Appendix A reveals that the miR-30a, miR-141, miR-200a, and miR-200b levels did not differ significantly.

### 3.5. Combination of the miRNAs and PTEN Panel to Predict SH/CH-nonA Progression to EC

Among the potential biomarkers, the miRNAs (miR-30a-3p, miR-141, miR-200a, and miR-200b) and PTEN status were included in panels for predicting future EC (Table 1). The combination of complete PTEN loss and miR-200a provided the optimal performance for predicting subsequent EC (sensitivity = 0.760; specificity = 1.000; PPV = 1.000; NPV = 0.937; accuracy = 0.947). Figure 5 provides a summarized study profile.

## 4. Discussion

We identified 49 patients with EC preceded by SH/CH-nonA (case) and excluded 23 of these; the SH/CH-nonA diagnoses of 15 of these 23 patients were revised to CAH, reflecting the limitations of the hematoxylin and eosin staining used in conventional pathology. Therefore, biomarkers must be identified for more accurate EH classification and EC risk analysis for patients with SH/CH-nonA. To the best of our knowledge, the present study is the first to investigate miRNA signatures with a focus on the differences between SH/CH-nonA that progresses or does not progress to EC, rather than the cross-sectional differences among cases of EC with CAH, SH, or normal endometrial tissue.

In the present study, miRNA array analyses revealed that 20 miRNAs were significantly different between the control and case patients. During the validation phase, multiplex RT-qPCR revealed that the levels of four of these 20 miRNAs (i.e., miR-30a-3p, miR-141, miR-200a, and miR-200b) were significantly higher in the control patients than in the case patients. We also performed a prediction of the biological processes targeted by the miRNAs by using the data from a website (https://fadeel.shinyapps.io/miRNA-GO-analysis, accessed on 16 January 2022) that provides a gene ontology of regulated miRNAs [25]. However, the prediction of the biological processes targeted by the 20 miRNAs (including the four validated miRNAs) produced nonsignificant results (adjusted *p* > 0.05; Appendix A).

Among the four potential miRNA biomarkers, miR-141, miR-200a, and miR-200b were upregulated in EC [24,26]. The finding of miR-200a upregulation is consistent with the results of a study that compared EC tissues with CAH, SH, and normal endometrial tissues from multiple women [14] and of a systematic review revealing miR-200a to be frequently upregulated in EC [13]. Therefore, miR-141, miR-200a, and miR-200b can be used to predict the risk of progression from SH/CH-nonA to CAH or EC. The AUC values for miR-30a-3p, miR-141, miR-200a, and miR-200b for discriminating case from control patients were 0.623, 0.754, 0.783, and 0.704, respectively, and the combination of complete PTEN loss with miR-200a produced the optimal performance for predicting subsequent EC (sensitivity = 0.760; specificity = 1.000; PPV = 1.000; NPV 0.937; accuracy = 0.947) in women with SH or CH-nonA. 

The miR-200 family comprises five miRNAs (i.e., miR-141, miR-200a, miR-200b, miR-200c, and miR-429) [27], and in our study, three of the four miRNAs exhibiting differences between groups belonged to the miR-200 family, which targets the tumor suppressor PTEN [28]. Mutter et al. demonstrated a significant difference between EC and endometrial intraepithelial neoplasia in terms of the percentage of PTEN mutations (83% vs. 55%) but not in the complete loss of PTEN (61% vs. 75%) [29]. In our study, the percentage of complete PTEN loss was significantly higher in the case group than in the control group (24.0% vs. 0%, *p <* 0.0001); furthermore, the EC tissues of the case group exhibited significantly more complete PTEN loss than their EH tissues did (56.0% vs. 24.0%, *p* = 0.001), whereas the levels of miR-141, miR-200a, and miR-200b were similar in the case EH and EC tissues. Collectively, the early miR-141, miR-200a, and miR-200b alterations led to progressive PTEN loss in the endometrial carcinogenesis pathway. The patients in our case group progressed to EC after a median time of 8.71 (range, 0.55–19.84) years.

We identified biomarkers potentially useful for identifying individuals with a high risk of EC. However, our study has several limitations. First, the sample of patients with SH/CH-nonA that progressed to EC was small; therefore, larger cohorts are required in future research. We plan to conduct a validation study involving an independent cohort and the use of a multicenter trial platform (Taiwanese Gynecologic Oncology Group). Second, our results indicate that miR-30a-3p is involved in the progression from SH/CH-nonA to EC. miR-30a-3p may act as an oncogene in ovarian cancer [30] and melanoma cells [31]. However, Tsukamoto et al. reported miR-30a-3p to be a negative regulator in EC [32]. The inconsistency between our findings and those of Tsukamoto et al. stems from the difference in patient selection. We selected only patients with SH/CH-nonA and compared those who developed (case) and did not develop (control) EC. By contrast, Tsukamoto et al. compared EC tissues with normal endometrial tissues because miR-30a-3p is involved in the negative regulation of BECN1 [33,34]. BECN1 is an essential protein for the initiation of autophagosome formation [35], and the antitumor effect of BECN1 is receiving increased attention [36]. However, in our study, the BECN1 was not related to the development of EC among patients with SH/CH-nonA. Third, because a single miRNA can target hundreds of mRNAs [37], miR-30a-3p expression was considered a candidate biomarker for EC carcinogenesis, such as through the targeting of E-cadherin [38] and cancer/testis antigen (CAGE) [31,39] by miR-30a-3p and promotion of cell cycle progression by miR-200b to regulate cyclin D1 [40]. Furthermore, cyclin D1 levels were demonstrated to differ significantly between EH and EC [41]. Therefore, further studies of other targets of miR-30a-3p and miR-200b are warranted. SH/CH-nonA diagnosis is highly subjective, with pathologists frequently disagreeing on the criteria for architectural crowding and nuclear atypia, which confer a significantly increased risk of progression to EC. No study has examined the application of miRNA and PTEN profiles for histopathological prediction of EC risk among patients with SH/CH-nonA. Research on the profiles of the aforementioned tissue biomarkers may help us identify which women with SH/CH-nonA have a high risk of EC.

## 5. Conclusions

Complete PTEN loss, miR-30a-3p, miR-141, miR-200a, and miR-200b are potentially useful tissue biomarkers of EC risk among patients with SH/CH-nonA.

## Figures and Tables

**Figure 1 jpm-12-00311-f001:**
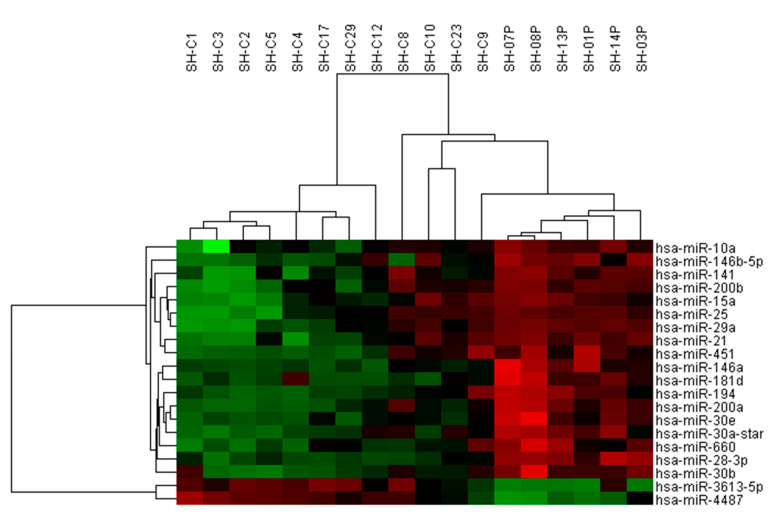
Among patients with SH/CH-nonA, the levels of 20 miRNAs were significantly higher in those who progressed to EC than in those who did not. Hierarchical clustering analysis of the miRNA profiles of the case group (SH-01P, SH-03P, SH-07P, SH-08P, SH-13P, and SH-14P) and control group (SH-C1, SH-C2, SH-C3, SH-C4, SH-C5, SH-08P, SH-C9, SH-C10, SH-12, SH-17, SH-23, and SH-29) was performed. Downregulated miRNAs are presented in green, and upregulated miRNAs are presented in red.

**Figure 2 jpm-12-00311-f002:**
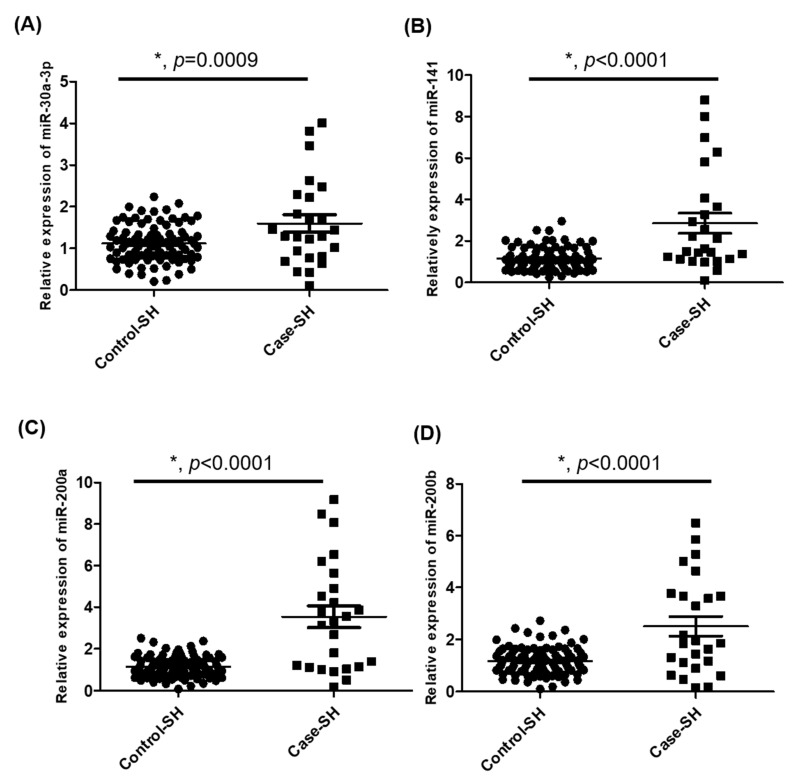
Differential expression of miRNAs in expanded cohort. (**A–D**) RT-qPCR results for miR-30a-3p, miR-141, miR-200a, and miR-200b levels, respectively, which were significantly higher in patients whose SH/CH-nonA progressed to EC (Case-SH) than in those whose did not (Control-SH). The expression of miRNAs was normalized to endogenous control miR-16. * *p <* 0.05.

**Figure 3 jpm-12-00311-f003:**
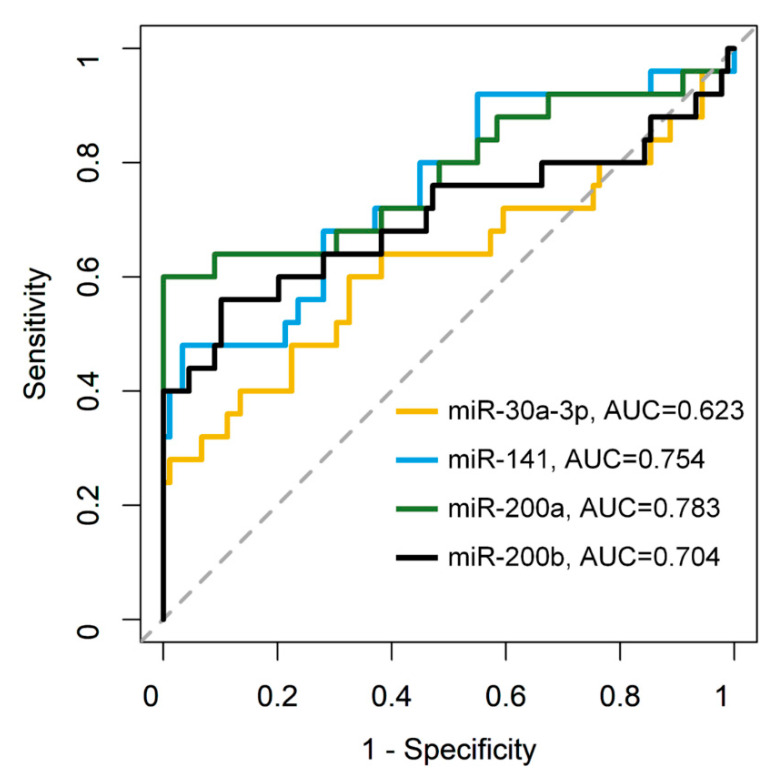
AUC analysis of miR-30a-3p, miR-141, miR-200a, and miR-200b.

**Figure 4 jpm-12-00311-f004:**
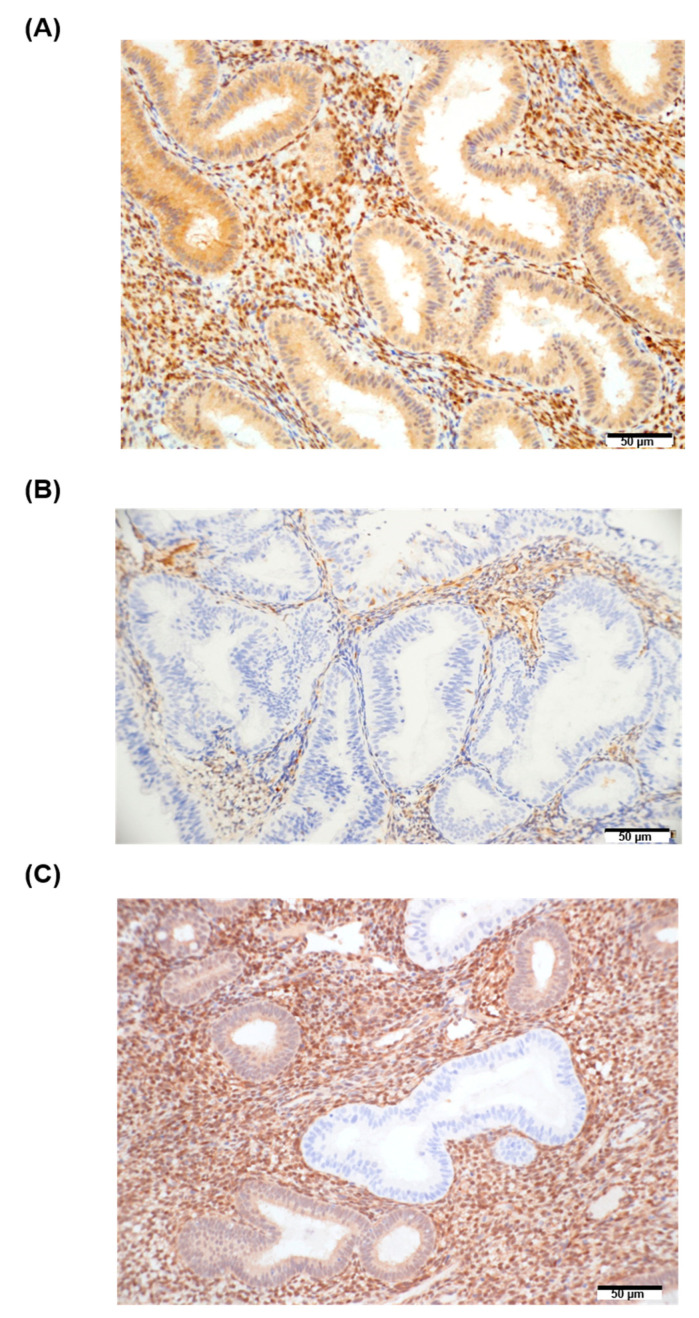
Degree of PTEN loss in endometrial hyperplasia without atypia: (**A**) Expression of PTEN in the cytoplasm of endometrial glandular epithelium. Control sample of simple hyperplasia (SH-C1); cancer did not develop subsequently. (**B**) Endometrial hyperplasia without atypia that preceded to EC (SH-06P); complete PTEN loss was observed. (**C**) Sample of endometrial hyperplasia without atypia (SH-C52); PTEN expression is heterogeneous. The positive staining in the cytoplasm of the endometrial stromal cells served as an internal positive control.

**Figure 5 jpm-12-00311-f005:**
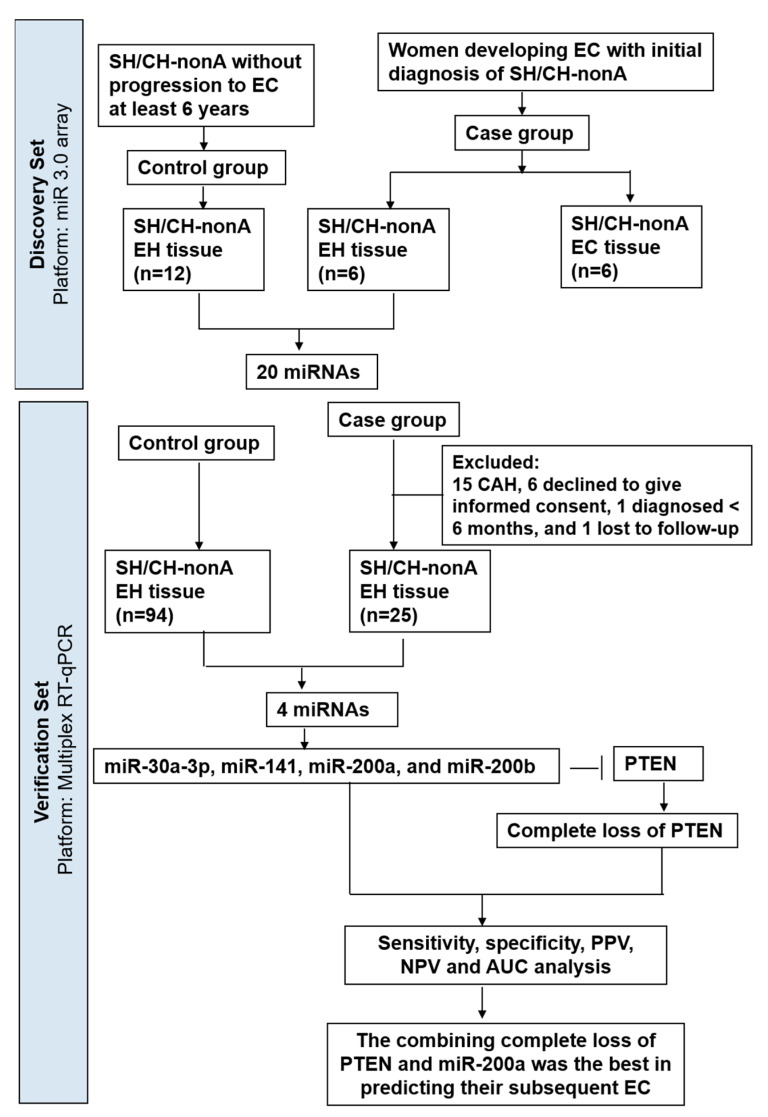
Study profile. During the discovery phase, miRNA array analyses revealed that 20 miRNAs differed significantly (*p <* 0.05, fold change >4) between the control (*n* = 12) and case (*n* = 6) patients. During the validation phase, multiplex RT-qPCR revealed that, among the 20 miRNAs, miR-30a-3p, miR-141, miR-200a, and miR-200b were significantly different (*p <* 0.05) between the expanded control (*n* = 94) and case (*n* = 25) cohorts. Sensitivity, specificity, PPV, NPV, and AUC analyses were performed to evaluate the performance of the four miRNAs in combination with miRNA-targeted complete PTEN loss for predicting EC risk among patients with SH/CH-nonA.

**Table 1 jpm-12-00311-t001:** Discriminative ability of various variables and panels for distinguishing patients whose SH/CH-nonA would subsequently progress to EC from those whose would not.

Variables	Sensitivity	Specificity	Accuracy
miR-30a-3p	0.600	0.674	0.658
miR-141	0.480	0.966	0.860
miR-200a	0.600	1.000	0.912
miR-200b	0.560	0.899	0.825
PTEN	0.520	1.000	0.895
miR-30a-3p + PTEN	0.500	0.989	0.878
miR-141 + PTEN	0.520	1.000	0.895
miR-200a + PTEN	0.760	1.000	0.947
miR-200b + PTEN	0.600	1.000	0.912

**Table 2 jpm-12-00311-t002:** PTEN IHC stain results of EH tissue in case and control patients with SH/CH-nonA and EC tissue in case patients.

Count (%)	PTEN Complete Loss	PTEN Heterogeneous	PTEN Positive	Total
Control-SH/CH-nonA	0 * (0%)	53 (56.4%)	41 (43.6%)	94 (100%)
Case-SH/CH-nonA	6 *^,#^ (24.0%)	10 (40.0%)	9 ^+^ (36.0%)	25 (100%)
Subsequent endometrial cancer of Case-SH/CH-nonA	14 ^#^ (56.0%)	11 (44.0%)	0 ^+^ (0%)	25 (100%)

* *p <* 0.001. ^#^
*p* = 0.001. ^+^
*p* = 0.002.

## Data Availability

The data presented in this study are available on request from the corresponding author.

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
