# Peer review of "MicroRNAs as Predictors of Future Uterine Malignancy in Endometrial Hyperplasia without Atypia"

_jpm, 2022, doi:10.3390/jpm12020311_

Round 1

Reviewer 1 Report

Although the authors explained the "limitations of the study" that this study is suffering from the small sample size, this is still a big issue and a main concern of the reviewer. This is an interesting study with remarkable results, however, the small sample size is not supported for the comprehensive conclusion. I am sorry that I cannot be positive and recommend this manuscript for publication.

Author Response

Reply: Thank you for your comments. We added a sentence “We are planning a validation study of an independent cohort through a multicenter trial group platform (Taiwanese Gynecologic Oncology Group).” in last paragraph before conclusion to address how we would like to overcome the limitation of small sample size. (line 315~316)

Reviewer 2 Report

This work deals with studying uRNAs being feasible predictors for diagnostics of uterine malignancy associated with endometrial hyperplasia.

The topic is highly actual regarding molecularly based early diagnostics of cancer progression.

In this manner, it does deserve an attention.

REMARKS

INTRODUCTION (note)

Introduction is well written - it deals with general presentation of EC, brief statistics in Taiwan, worldwide classification of EH, description of roles and achievements in analysis of uRNAs being feasible biomarkers for early cancer diagnostics and eventually authors aim for suggested work.

However, I miss to add a general statement of biomarkers´ roles in cancer and board diagnostics of various diseases. Please, follow the down below upgrade.

INTRODUCTION (upgrade)

„The trend in searching for biomarkers for early diagnostics of cancers is growing [https://doi.org/10.1002/pmic.202100198]. As microRNAs are small RNA and well preserved in formalin, there is also interest in searching for microRNA biomarkers and corresponding potential target genes of transitions from SH/CH-nonA to EC with longer duration of follow-up.“

MATERIALS AND METHODS

TISSUE SPECIMENS (notes)

Consider if it would be better to use term “tissue samples” instead of “tissue specimens”.

Please, insert here declaration of “ethical approval” for this experiment with number.

RNA EXTRACTION FROM FFPE (notes)

Please, also provide countries (besides of the name of products) of reagents/instruments used in this protocol.

Provide the “http” in brackets where to find mentioned manufactured instructions or reference if possible.

Please, specify extraction mode to isolate uRNAs from FFPE.

I would very appreciate if you would provide a scheme depiction of the whole workflow as a figure involving steps such as database search, patients enrolling, sampling, sample preparation, detection, and assessment via bioinformatics engines!!! This would certainly help for the reader to quickly catch the whole process of investigatioin.

uRNA 3.0 ARRAY AND TARGET PREDICTION OF uRNAs (note)

Please provide a reference (http) where can be find manufacturer´s instructions.

QUANTITATIVE REAL-TIME PCR (note)

Provide information of country of origin of used uRNA Assay approach.

RESULTS (note)

Results are exhaustive.

Author Response

This work deals with studying uRNAs being feasible predictors for diagnostics of uterine malignancy associated with endometrial hyperplasia.

The topic is highly actual regarding molecularly based early diagnostics of cancer progression.

In this manner, it does deserve an attention.

REMARKS

INTRODUCTION (note)

Introduction is well written - it deals with general presentation of EC, brief statistics in Taiwan, worldwide classification of EH, description of roles and achievements in analysis of uRNAs being feasible biomarkers for early cancer diagnostics and eventually authors aim for suggested work.

However, I miss to add a general statement of biomarkers´ roles in cancer and board diagnostics of various diseases. Please, follow the down below upgrade.

INTRODUCTION (upgrade)

“The trend in searching for biomarkers for early diagnostics of cancers is growing [https://doi.org/10.1002/pmic.202100198]. As microRNAs are small RNA and well preserved in formalin, there is also interest in searching for microRNA biomarkers and corresponding potential target genes of transitions from SH/CH-nonA to EC with longer duration of follow-up.“

Reply: Thank you for your suggestion, we have added “The trend in searching for biomarkers for early diagnostics of cancers is growing [https://doi.org/10.1002/pmic.202100198]” in introduction (line 61) (new reference 6).

MATERIALS AND METHODS

TISSUE SPECIMENS (notes)

Consider if it would be better to use term “tissue samples” instead of “tissue specimens”.

Please, insert here declaration of “ethical approval” for this experiment with number.

Reply: Thank you for your suggestion, we have used term “tissue samples” instead of “tissue specimens” (line 94) and insert “This study was approved by the Institutional Review Board of the Chang Gung Memorial Hospital, Taiwan ((IRB approval numbers: 100-4355A3 and 201700921B0)” (line 95~96)

RNA EXTRACTION FROM FFPE (notes)

Please, also provide countries (besides of the name of products) of reagents/instruments used in this protocol.

Provide the “http” in brackets where to find mentioned manufactured instructions or reference if possible.

Please, specify extraction mode to isolate uRNAs from FFPE.

Reply: Thank you for your suggestion, we have provided the city and the county of reagents/instruments used in this protocol. (line 112 and 116) and provided the reference in brackets where to find mentioned manufactured instructions (new reference 18). In addition, we have specified extraction mode “In brief, sample lysis was digested with proteinase K at 56°C for 15 min, and treated with DNase at room temperature for 15 min. RNA was purified through filter cartridges” in RNA extraction mode to isolate RNA from FFPE (line 114~115).

I would very appreciate if you would provide a scheme depiction of the whole workflow as a figure involving steps such as database search, patients enrolling, sampling, sample preparation, detection, and assessment via bioinformatics engines!!! This would certainly help for the reader to quickly catch the whole process of investigation.

Reply: Thank you for your suggestion, we have added Figure 5 for study profile. In discovery phase, 20 miRNAs (p < 0.05, fold change > 4) were significantly different in SH/CH-nonA controls (n = 12) compared to SH/CH-nonA cases (n = 6) by miRNA array. In validation phase, among 20 miRNAs, 4 miRNAs for miR30a-3p, miR141, miR200a and miR200b (p < 0.05) were significantly different in SH/CH-nonA controls (n = 94) compared to SH/CH-nonA cases (n = 25) by multiplex RT-qPCR platform. The sensitivity, specificity, PPV, NPV and AUC were performed to evaluate the performance of the 4 miRNAs with miR-targeted PTEN complete loss for the detection for predicting future risk of developing EC for patients with SH/CH-nonA. (line 261~268).

uRNA 3.0 ARRAY AND TARGET PREDICTION OF uRNAs (note)

Please provide a reference (http) where can be find manufacturer´s instructions.

Reply: We have provided new reference 20 and 21 for find manufacturer´s instructions in miRNA 3.0 array and target prediction of miRNAs section. (line 123)

QUANTITATIVE REAL-TIME PCR (note)

Provide information of country of origin of used uRNA Assay approach.

Reply: Thank you for your suggestion, we have provided the city and the county of reagents/instruments used in this QUANTITATIVE REAL-TIME PCR. (line 142)

RESULTS (note)

Results are exhaustive.

Reply: We deeply appreciated your comments.

Round 2

Reviewer 1 Report

For me, the author's argument is not convincing and I cannot recommend this MS for publication.

Reviewer 2 Report

Authors have responded to queries given. Thus, I consider acceptation of the manuscript at current form.

This manuscript is a resubmission of an earlier submission. The following is a list of the peer review reports and author responses from that submission.

Round 1

Reviewer 1 Report

Thank you for the opportunity to review this article. 

This study aimed to investigate the molecular biomarkers for risk of developing endometrial cancer (EC) in women with simple/complex hyperplasia without atypia (SH/CH-nonA). EH tissues of EC patients with preceding SH/CH-nonA (case; n=6) were compared to SH/CH-nonA patients without progression to EC (control; n=12) using miRNA array, verified in an independent cohort (case, n=25; and control; n=94) by RT-qPCR. As the results, 20 significantly different miRNAs were identified by miRNA array. Multiplex RT-qPCR validation selected miR30a-3p, miR141, miR200a, and miR200b. Area under the curve values for tissue miR30a-3p, miR141, miR200a, miR200b in predicting future ECs were 0.623, 0.754, 0.783, and 0.704. A combination of complete PTEN loss with miR200a was the best in prediction performance.

Interesting topic. But this study struggles with the definition of the study population and small sample size.

  1. First of all, there were only 6 and 25 cases for miRNA array and RT-qPCR, respectively.
  2. For the case group, the authors selected patients with history of endometrial SH/CH-nonA subsequently developed EC (> 6 months from EH to EC). Meanwhile, patients proven to be SH/CH-nonA without progression to EC at least six years. I wonder how many patients received hormone therapy for what duration prior to the diagnosis of EC, which might affect expression or appearance of specific miRNAs.
  3. According to Supplementary Table 1, it seems that not all patients had endometrioid histologic subtype, which is also problematic for such a small study population.
  4. Lastly, I wonder prediction performance of PTEN loss itself.
  5. Baseline characteristics of the patients should be presented well (both for miRNA array RT-qPCR).
  6. The authors should mention limitations of the study in Discussion section.

Author Response

This study aimed to investigate the molecular biomarkers for risk of developing endometrial cancer (EC) in women with simple/complex hyperplasia without atypia (SH/CH-nonA). EH tissues of EC patients with preceding SH/CH-nonA (case; n=6) were compared to SH/CH-nonA patients without progression to EC (control; n=12) using miRNA array, verified in an independent cohort (case, n=25; and control; n=94) by RT-qPCR. As the results, 20 significantly different miRNAs were identified by miRNA array. Multiplex RT-qPCR validation selected miR30a-3p, miR141, miR200a, and miR200b. Area under the curve values for tissue miR30a-3p, miR141, miR200a, miR200b in predicting future ECs were 0.623, 0.754, 0.783, and 0.704. A combination of complete PTEN loss with miR200a was the best in prediction performance.

Interesting topic. But this study struggles with the definition of the study population and small sample size.

1. First of all, there were only 6 and 25 cases for miRNA array and RT-qPCR, respectively.

Reply:

A lot of patients with SH/CH-nonA were lost to follow-up or tissue blocks were missing, therefore only 6 and 25 cases had adequate FFPE tissues for miRNA array and RT-qPCR, respectively. We have mentioned limitations of the study in Discussion section. (line278~280)

First, we apologize for the miss-leading presentation. Since our study is limited by the small sample of patients with SH/CH-nonA cases indeed progressed to EC, and, for that reason, we sought confirmation of our findings in an expanded cohort consisting 25 cases had adequate FFPE tissues for RT-qPCR. We instead the “independent: cohort with “expanded” cohort. (line26, 82, 177, 180, 189)

Besides, the subject (SH-014P) who had MiRNA array but did not have adequate tissue for QPCR is added in Table S1.

2. For the case group, the authors selected patients with history of endometrial SH/CH-nonA subsequently developed EC (> 6 months from EH to EC). Meanwhile, patients proven to be SH/CH-nonA without progression to EC at least six years. I wonder how many patients received hormone therapy for what duration prior to the diagnosis of EC, which might affect expression or appearance of specific miRNAs.

Reply:

According to electronic medical records, there are 23 patients received hormone therapy from endometrial SH/CH-nonA to EC. After calculating, patients with received hormone therapy or not did not affect expression or appearance of specific miRNAs.

3. According to Supplementary Table 1, it seems that not all patients had endometrioid histologic subtype, which is also problematic for such a small study population.

Reply:

All cases were endometrioid histologic subtype, the two with adenocarcinoma on the original pathology report were confirmed to be endometrioid, therefore Supplementary Table 1 is revised.

4. Lastly, I wonder prediction performance of PTEN loss itself.

Reply:

The sensitivity, specificity, and accuracy data are added to Table 2.

5. Baseline characteristics of the patients should be presented well (both for miRNA array RT-qPCR).

Reply:

The 6 patients with miRNA are added in Table S1.

6. The authors should mention limitations of the study in Discussion section.

Reply: Thank you for your suggestion, we have added limitations of the study in Discussion section. (line278~280)

Reviewer 2 Report

This is study is suffering from three things from my point of view;

1- The introduction section is poorly written and is not informative. Many studies have been done in this domain but the authors just mentioned one single study in the introduction section; Lee's study, Ref No. 9. I highly recommend improving the introduction section by adding more information/data from previous studies.

2- The number of samples (6 vs 12) statistically is not big enough for the comprehensive conclusion.

3- There are some systematic review studies, that I recommend the authors to compare their results with these studies, e.g., https://www.ncbi.nlm.nih.gov/pmc/articles/PMC7260115/

Author Response

This is study is suffering from three things from my point of view;

1- The introduction section is poorly written and is not informative. Many studies have been done in this domain but the authors just mentioned one single study in the introduction section; Lee's study, Ref No. 9. I highly recommend improving the introduction section by adding more information/data from previous studies.

Reply:

(1) To our knowledge, our study is the first to investigate miRNA signatures focused on the differences of simple/complex hyperplasia without atypia (SH/CH-nonA) case (progression) versus control (without progression) rather than cross sectional comparisons between EC with CAH, SH and normal endometrial tissue.

(2) We add two articles (new refs. 11 and 12) about the molecular biomarkers for risk of developing EC in women with SH/CH-nonA rather than CAH and add some references in introduction section, such as: , Tantbirojn et al., reported that absence of PTEN protein expression was detected in 60% of EC, and 24% of typical endometrial hyperplasia (SH/CH-nonA) [11]. Sangwan et al., reported that the expression of cyclin D1 was significantly increased in EC compared with SH [12]. (line74~78)

(3) We add in discussion Xia et al., found that miR200b promoted cell cycle progression by regulating cyclin D1 [32] and there was a statistically significant difference in cyclin D1 expression between SH and EC. We have added in discussion section. (line293~297)

2- The number of samples (6 vs 12) statistically is not big enough for the comprehensive conclusion.

Reply:

A lot of patients with SH/CH-nonA were lost to follow-up or tissue blocks were missing, therefore only 6 and 25 cases had adequate FFPE tissues for miRNA array and RT-qPCR, respectively. We have mentioned limitations of the study in Discussion section. (line278~280)

First, we apologize for the miss-leading presentation. Since our study is limited by the small sample of patients with SH/CH-nonA cases indeed progressed to EC, and, for that reason, we sought confirmation of our findings in an expanded cohort consisting 25 cases had adequate FFPE tissues for RT-qPCR. We instead the “independent: cohort with “expanded” cohort. (line 26, 82, 177, 180, 189)

Besides, the subject (SH-014P) who had MiRNA array but did not have adequate tissue for QPCR is added in Table S1.

3- There are some systematic review studies, that I recommend the authors to compare their results with these studies, e.g., https://www.ncbi.nlm.nih.gov/pmc/articles/PMC7260115/

Reply:

Thank you for your suggestion, we have added this systematic review studies in discussion section as refs 18 and 29. (line 254~256, line 291)

Round 2

Reviewer 1 Report

This study struggles with the definition of the study population and small sample size. 

Author Response

This study struggles with the definition of the study population and small sample size. 

Reply:

Yes, we have admitted that our study is limited by the small sample of patients with SH/CH-nonA cases actually progressed to EC. A total of 49 cases EC patients with preceding SH/CH-nonA (case) were identified, of which 23 cases we excluded for analysis (15 CAH, 6 declined to give informed consent, 1 diagnosed < 6 months, and 1 lost to follow-up). The fact that 15 of the 23 excluded case had their diagnosis from SH/CH-nonA revised to CAH reflected the limitations of H & E stain of conventional pathology. Therefore, finding potential biomarkers for more accurate classification of EH and predicting future risk of developing EC for patients with SH/CH-nonA is necessary. Such information was added in abstract (line 24-26), section 2.1. (line 95-97), and discussion (line 257-260).

Since our study is limited by the small sample of patients with SH/CH-nonA cases indeed progressed to EC, and, for that reason larger cohorts are needed. We have added limitations of the study in discussion section (line 294~296)

We have also modified our sentences that our study suggests that complete loss of PTEN, miR30a-3p, miR141, miR-200a, and miR200b are potentially useful tissue biomarkers for predicting future risk of developing EC for patients with SH/CH-nonA. (line 39, 292, 319)

Reviewer 2 Report

My comments are the same as before. The authors did not address my concerns properly and did not take them seriously. 

This is study is suffering from three things from my point of view;

1- The introduction section is poorly written and is not informative. Many studies have been done in this domain but the authors just mentioned one single study in the introduction section; Lee's study, Ref No. 9. I highly recommend improving the introduction section by adding more information/data from previous studies.

2- The number of samples (6 vs 12) statistically is not big enough for the comprehensive conclusion.

3- There are some systematic review studies, that I recommend the authors to compare their results with these studies, e.g., https://www.ncbi.nlm.nih.gov/pmc/articles/PMC7260115/

The figures are also blurry and need to be revised.

Author Response

My comments are the same as before. The authors did not address my concerns properly and did not take them seriously. 

Reply:

I am sorry that I do not address your suggestions properly. We have rewritten and reorganized the information as below.

This is study is suffering from three things from my point of view;

  • The introduction section is poorly written and is not informative. Many studies have been done in this domain but the authors just mentioned one single study in the introduction section; Lee's study, Ref No. 9. I highly recommend improving the introduction section by adding more information/data from previous studies.

Reply:

Recently, Donkers et al., [11] summarized that miR205, the miR200 family, miR135b, miR182, miR183 and miR223 were potentially biomarker for the diagnosis of EC and depicted cases of atypical hyperplasia prone to faster progression into EC [12,13]. Moreover, Tantbirojn et al., [14] reported that absence of PTEN protein expression was detected in 60% of EC, and 24% of typical endometrial hyperplasia (SH/CH-nonA). Sangwan et al., [15] reported that the expression of cyclin D1 was significantly increased in EC compared with SH. However, microRNA profiling can reveal early diagnostic clues of future EC at the time of SH/CH-nonA which has not been fully investigated. We have added in introduction section (line 77~84)

  • The number of samples (6 vs 12) statistically is not big enough for the comprehensive conclusion.

Reply:

Yes, we have admitted that our study is limited by the small sample of patients with SH/CH-nonA cases actually progressed to EC. A total of 49 cases EC patients with preceding SH/CH-nonA (case) were identified, of which 23 cases we excluded for analysis (15 CAH, 6 declined to give informed consent, 1 diagnosed < 6 months, and 1 lost to follow-up). The fact that 15 of the 23 excluded case had their diagnosis from SH/CH-nonA revised to CAH reflected the limitations of H & E stain of conventional pathology. Therefore, finding potential biomarkers for more accurate classification of EH and predicting future risk of developing EC for patients with SH/CH-nonA is necessary. Such information was added in abstract (line 24-26), section 2.1. (line 95-97), and discussion (line 257-260).

Since our study is limited by the small sample of patients with SH/CH-nonA cases indeed progressed to EC, and, for that reason larger cohorts are needed. We have added limitations of the study in discussion section (line 294~296)

We have also modified our sentences that our study suggests that complete loss of PTEN, miR30a-3p, miR141, miR-200a, and miR200b are potentially useful tissue biomarkers for predicting future risk of developing EC for patients with SH/CH-nonA. (line 39, 292, 319)

3- There are some systematic review studies, that I recommend the authors to compare their results with these studies, e.g., https://www.ncbi.nlm.nih.gov/pmc/articles/PMC7260115/

Reply:

Thank you for your comments. We have rewritten and added more information such as: a recent systematic review by Donkers et al., found miR205, miR200c, miR223, miR182, miR183 and miR200a were frequently upregulated in EC [11]. Moreover, we also compared that miR141, miR200a and miR200b were upregulated in line with previously lecture reported in EC [12,13,21] as well as miR200b has been reported that promotes cell cycle progression by regulating cyclin D1 [35] and there was a statistically significant difference in cyclin D1 expression between SH and EC. We have added in introduction and discussion section (line 77~84, line 268~274, line 309~313)

The figures are also blurry and need to be revised.

Reply: Thank you very much. We have modified the pictures of PTEN in Figure 4 (page 9)